# PareCO: Pareto-aware Channel Optimization for Slimmable Neural Networks

## Abstract

Slimmable neural networks provide a flexible trade-off front between prediction error and computational cost (such as the number of floating-point operations or FLOPs) with the same storage cost as a single model. They have been proposed recently for resource-constrained settings such as mobile devices. However, current slimmable neural networks use a single width-multiplier for all the layers to arrive at sub-networks with different performance profiles, which neglects that different layers affect the network's prediction accuracy differently and have different FLOP requirements. Hence, developing a principled approach for deciding width-multipliers across different layers could potentially improve the performance of slimmable networks. To allow for heterogeneous width-multipliers across different layers, we formulate the problem of optimizing slimmable networks from a multi-objective optimization lens, which leads to a novel algorithm for optimizing both the shared weights and the width-multipliers for the sub-networks. We perform extensive empirical analysis with 15 network and dataset combinations and two types of cost objectives, *i.e.*, FLOPs and memory footprint, to demonstrate the effectiveness of the proposed method compared to existing alternatives. Quantitatively, improvements up to 1.7% and 8% in top-1 accuracy on the ImageNet dataset can be attained for MobileNetV2 considering FLOPs and memory footprint, respectively. Our results highlight the potential of optimizing the channel counts for different layers jointly with the weights for slimmable networks.

## 1 Introduction

Slimmable neural networks have been proposed with the promise of enabling multiple neural networks with different trade-offs between prediction error and the number of floating-point operations (FLOPs), *all at the storage cost of only a single neural network* (Yu et al., 2019). This is in stark contrast to channel pruning methods (Berman et al., 2020; Yu & Huang, 2019a; Guo et al., 2020; Molchanov et al., 2019) that aim for a small standalone model. Slimmable neural networks are useful for applications on mobile and other resource-constrained devices. As an example, the ability to deploy multiple versions of the same neural network would alleviate the maintenance costs for applications which support a number of different mobile devices with different memory and storage constraints, as only one model needs to be maintained. Similarly, one can deploy a single model which is configurable at run-time to dynamically cope with different latency or accuracy requirements. For example, users may care more about power efficiency when the battery of their devices is running low while the accuracy of the ConvNet-powered application may be more important otherwise.

A slimmable neural network is trained by simultaneously considering networks with different widths (or channel counts) using a single set of shared weights. The width of a child network is specified by a real number between 0 and 1, which is known as the "width-multiplier" (Howard et al., 2017). Such a parameter specifies how many channels per layer to use proportional to the full network. For example, a width-multiplier of $0.35\times$ represents a network that has channel counts that are 35% of the full network for all the layers. While specifying child networks using a single width-multiplier for all the layers has shown empirical success (Yu & Huang, 2019b; Yu et al., 2019), such a specification neglects that different layers affect the network's output differently (Zhang et al., 2019) and have different FLOP requirements (Gordon et al., 2018), which may lead to sub-optimal results. In a similar setting, as demonstrated in the model pruning literature (Gordon et al., 2018; Liu et al., 2019b; Morcos et al., 2019; Renda et al., 2020), having different pruning ratios for different layers of

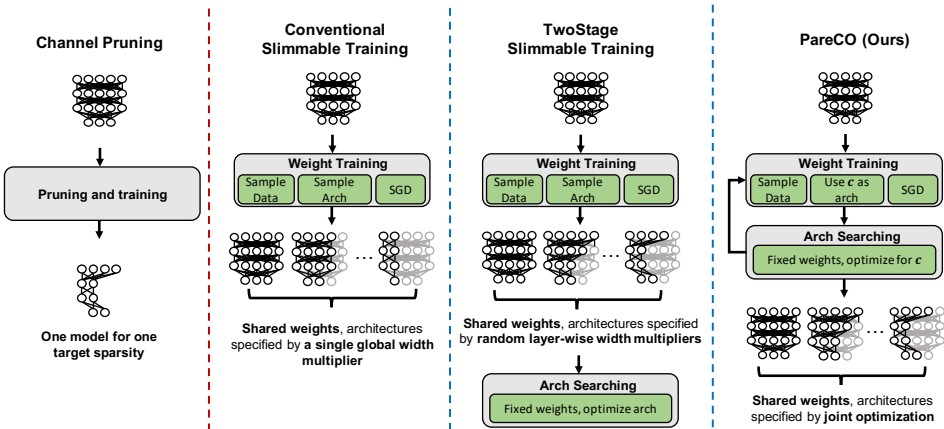

Figure 1: Schematic overview comparing our proposed method with existing alternatives and channel pruning. Channel pruning has a fundamentally different goal compared to ours, *i.e.*, training slimmable nets. PareCO jointly optimizes both the architectures and the shared weights.

the network can further improve results over a single ratio across layers. This raises an interesting question: *How should we obtain these non-uniform widths for slimmable nets?*

To achieve non-uniform width-multipliers across layers, one can consider using techniques from the neural architecture search (NAS) literature (Cai et al., 2020; Yu et al., 2020), which we call *TwoStage* training. Specifically, one can first train a supernet with weight-sharing by uniformly sampling width-multiplier for each layer. After this procedure converges, one can use multi-objective optimization methods to search for width given the trained weights. However, width optimization has a much larger design space than that considered in existing methods for NAS. Specifically, each layer can have hundreds of choices (since there are hundreds of channels for each layer). This makes it unclear if such a training technique is suitable for channel optimization[1].

As an alternative to existing techniques, we take a multi-objective optimization viewpoint, aiming to jointly optimize the width-multipliers for different layers and the shared weights in a slimmable neural network. A schematic view of the differences among the conventional slimmable training, TwoStage training, and our proposed method is shown in Figure 1. The contributions of this work are three-fold. First, through a multi-objective optimization lens, we provide the first principled formulation for jointly optimizing the weights and widths of slimmable neural networks. The proposed formulation is general and can be applied to objectives other than prediction error and FLOPs (Yu & Huang, 2019b; Yu et al., 2019). Second, we propose Pareto-aware Channel Optimization or PareCO, a novel algorithm which approaches the intractable problem formulation in an approximate fashion using stochastic gradient descent, of which the conventional training method proposed for universally slimmable neural networks (Yu & Huang, 2019b) is a special case. Finally, we perform extensive empirical analysis using 15 network and dataset combinations and two types of cost objectives to demonstrate the effectiveness of the proposed algorithm over existing techniques.

## 2 RELATED WORK

### 2.1 SLIMMABLE NEURAL NETWORKS

Slimmable neural networks (Yu et al., 2019) enable multiple sub-networks with different compression ratios to be generated from a single network with one set of weights. This allows the FLOPs of network to be dynamically configurable at run-time without increasing the storage cost of the model weights. Based on this concept, better training methodologies have been proposed to enhance the performance of slimmable networks (Yu & Huang, 2019b). One can view a slimmable network as a dynamic

---

[1]Both OFA (Cai et al., 2020) and BigNAS (Yu et al., 2020) mainly use the pre-defend channel counts and search for kernel sizes, depth, and input resolution. Specifically, channel counts refer to expansion ratios only for OFA while BigNAS only considers a small range of channel counts near the pre-defined ones.

computation graph where the graph can be constructed dynamically with different accuracy and FLOPs profiles. With this perspective, one can go beyond changing just the width of the network. For example, one can alter the network's sub-graphs (Ruiz & Verbeek, 2019), network's depth (Bolukbasi et al., 2017; Elbayad et al., 2019; Huang et al., 2017; Li et al., 2019a; Kaya et al., 2019), and network's kernel sizes and input resolutions (Cai et al., 2020; Yu et al., 2020). Complementing prior work primarily focusing on generalizing slimmable networks to additional architectural paradigms, our work provides the first principled multi-objective formulation for optimizing slimmable networks with tunable architecture decisions. While our analysis focuses on the network widths, our proposed formulation can be easily extended to other architectural parameters.

## 2.2 NEURAL ARCHITECTURE SEARCH

A slimmable neural network can be viewed as an instantiation of weight-sharing. In the literature for neural architecture search (NAS), weight-sharing is commonly adopted to reduce the search cost (Liu et al., 2018; Stamoulis et al., 2019; Guo et al., 2019; Bender et al., 2018; Berman et al., 2020; Yu & Huang, 2019a). Specifically, NAS methods use weight-sharing as a proxy for evaluating the performance of the sub-networks to reduce the computational cost of iterative training and evaluation. However, NAS methods are concerned with the architecture of the network and the found network is re-trained from scratch, which is different from the weight-sharing mechanism adopted in slimmable networks where the weights are used for multiple networks during test time.

Multi-objective optimization has also been adopted in NAS literature (Dong et al., 2018; Cheng et al., 2018; Iqbal et al., 2020; Lu et al., 2019; Elsken et al., 2018). However, a crucial difference of the present work compared to these papers is that we are interested in learning a single set of weights from which multiple FLOP configurations can be used (as in slimmable networks) rather than finding architectures independently for each FLOP configuration that can be trained from scratch freely. Put another way, in our setting, both the shared weights and the searched architectures optimized jointly, whereas in prior work, only searched architectures were optimized.

## 2.3 CHANNEL PRUNING

Reducing the channel or filter counts for a pre-trained model is also known as channel pruning. In channel pruning, the goal is to find a single small model that maximizes the accuracy while satisfying some resource constraints. Several studies have investigated how to better characterize redundant channels in a post-training fashion given a pre-trained model (Li et al., 2016; He et al., 2019; Liu et al., 2017; Ye et al., 2018; Molchanov et al., 2016; 2019; Aflalo et al., 2020). Besides a post-processing perspective to channel pruning, prior work has also investigated channel pruning via an optimization lens. Specifically, channel pruning methods based on Lasso (Wen et al., 2016; Liu et al., 2017; Gordon et al., 2018; Yun et al., 2019), stochastic $\ell_0$ (Louizos et al., 2017), and ADMM (Li et al., 2019b; Yang et al., 2020) have been developed. Liu *et al.* (Liu et al., 2019b) later show that channel counts for different layers are more important for the performance of channel pruning. As a result, several studies have investigated pruning via an architecture search perspective (Yu & Huang, 2019a; Berman et al., 2020; Ma et al., 2019; Chin et al., 2020; Liu et al., 2019a).

While channel pruning also optimizes for non-uniform widths, the goal of channel pruning is crucially different from ours. The key difference is that channel pruning is concerned with a single pruned model while slimmable neural networks require a set of models to be trained using weight sharing. As a result, it is not clear if those techniques are suitable for finding non-uniform widths for slimmable nets. Moreover, using channel pruning to obtain a set of pruned models can be computationally expensive. Specifically, the state-of-the-art differentiable pruning method takes roughly $2\times$ the training time for searching the channel counts for a given target FLOPs (Guo et al., 2020). That is, if we want 20 uniformly distributed FLOPs between 20% and 100%, it takes roughly $24\times$ the training time of the 100% FLOPs network just to obtain architectures. In contrast, our formulation is targeting slimmable nets directly with a training cost similar to that of conventional slimmable networks, which is $4\times$ the training time of a standalone model.

## 3 METHODOLOGY

In this work, we are interested in jointly optimizing the network widths and network weights. Ultimately, when evaluating the performance of a slimmable neural network, we care about the trade-off curve between multiple objectives, *e.g.*, theoretical speedup and accuracy. This trade-off curve is formed by evaluating multiple networks with width configurations sampled from a width sampling distribution. Viewed from this perspective, the sampling distribution should be optimized in such a way that the resulting networks have a better trade-off curve (*i.e.*, larger area under curve), which is a multi-objective optimization problem. This section formalizes this idea and provides an algorithm to solve it in an approximate fashion.

### 3.1 PROBLEM FORMULATION

Intuitively, our goal is to optimize the shared weights to maximize the area under the best trade-off curve between the accuracy and theoretical speedup obtained by optimizing network's widths. Since accuracy is not differentiable w.r.t. the shared weights, we switch objectives from accuracy and theoretical speedup to cross-entropy loss and FLOPs, respectively. In this setting, the objective becomes to *minimize* the area under curve. To arrive at such an objective, we start by defining the notion of optimality in *minimizing* multiple objectives (such as the cross-entropy loss and FLOPs).

**Definition 1 (Pareto frontier)** *Let $\boldsymbol{f}(\boldsymbol{x}) = (f_1(\boldsymbol{x}), \ldots, f_K(\boldsymbol{x}))$ be a vector of responses from $K$ different objectives. Define vector inequality $\boldsymbol{x} < \boldsymbol{y}$ as $x_i \leq y_i \,\forall\, i \in [K]$ with at least one inequality being strict. We call a set of points $\mathcal{P}$ a Pareto frontier if $\boldsymbol{f}(\boldsymbol{x}) < \boldsymbol{f}(\boldsymbol{y})$, for any $\boldsymbol{x} \in \mathcal{P}$ and $\boldsymbol{y} \notin \mathcal{P}$.*

With this definition, we essentially want the loss for the shared weights to be the area under the curve formed by the Pareto frontier. To do so, we need an actionable way to obtain the Pareto frontier and we make use of the following Lemma:

**Lemma 3.1 (Augmented Tchebyshev Scalarization (Section 1.3.3 in (Nakayama et al., 2009)))** *Define a scalarization of $K$ objectives as*

$$\mathcal{T}_{\boldsymbol{\lambda}}(\boldsymbol{x}) = \max_{i \in [K]} \lambda_i (f_i(\boldsymbol{x}) - \bar{f}_i) + \beta \sum_{i \in [K]} \lambda_i f_i(\boldsymbol{x}), \tag{1}$$

*where $\boldsymbol{\lambda}$ is weightings among objectives, $\bar{f}_i$ is a baseline constant such that $(f_i(\boldsymbol{x}) - \bar{f}_i) \geq 0 \,\forall\, \boldsymbol{x}$, and $\beta > 0$, the Pareto frontier can be specified via $\mathcal{P} = \{\arg\min_{\boldsymbol{x}} \mathcal{T}_{\boldsymbol{\lambda}}(\boldsymbol{x}) \,\forall\, \boldsymbol{\lambda} \in \Delta^{K-1}\}$ where $\Delta^{K-1}$ is a K-1 simplex.*

The second term of equation (1) is the commonly used weighted sum scalarization and it can be depicted as a line in the objectives space. However, minimizing it alone is not sufficient if the Pareto curve is non-convex (Nakayama et al., 2009). This calls for the first term, which can be depicted as the axis-aligned lines of a non-positive orthant in the objectives space. With Lemma 3.1, one can obtain the Pareto frontier by solving multiple augmented Tchebyshev scalarized optimization problems with different $\boldsymbol{\lambda}$s. A $\boldsymbol{\lambda}$ vector can be interpreted as a weighting on the objectives, which is used to summarize multiple objectives into a single scalar. For instance, consider the case in which the cross-entropy loss and FLOPs are the two objectives of interest. If taking $\lambda_{\text{CE}} \to 1$ and $\lambda_{\text{FLOPs}} \to 0$, the scalarized objective is then dominated by the cross-entropy loss and we are effectively seeking width configurations that minimize the cross entropy loss. In contrast, if taking $\lambda_{\text{CE}} \to 0$ and $\lambda_{\text{FLOPs}} \to 1$, we are then effectively seeking width configurations that minimize FLOPs. With the scalarization, we have the following theorem for summarizing the area under curve quantitatively:

**Theorem 3.2** *Optimizing network weights $\boldsymbol{\theta}$ to minimize area under Pareto curve formed by cross-entropy loss ($f_{CE}$) and FLOPs ($f_{FLOPs}$) can be done approximately with the following objective:*

$$\arg\min_{\boldsymbol{\theta}} \frac{1}{M} \sum_{m=1}^{M} f_{CE}(\alpha(\boldsymbol{g}(\boldsymbol{z}^{(\boldsymbol{m})})), \boldsymbol{\theta}, \boldsymbol{x}, y), \tag{2}$$

*where $\boldsymbol{x}, y$ are the sampled training input and label, respectively; $M$ is the number of Monte Carlo sample; $\boldsymbol{z}^{(\boldsymbol{m})}$ is a FLOPs target sampled uniformly between the lowest and the highest FLOPs, $\alpha(\cdot)$*

*maps a weighting $\boldsymbol{\lambda}$ to the corresponding Pareto channel configuration (i.e., $\alpha(\boldsymbol{\lambda}) = \arg\min_{\boldsymbol{c}} \mathcal{T}_{\boldsymbol{\lambda}}(\boldsymbol{c})$ where $\mathcal{T}$ and $\boldsymbol{\lambda}$ are defined in Lemma 3.1), and $g(\cdot)$ maps a certain FLOPs back to the corresponding $\boldsymbol{\lambda}$, (i.e., $g = (f_{FLOPs} \circ \alpha)^{-1}$).*

The proof is in Appendix A. Intuitively speaking, the objective is to minimize the cross entropy loss for the Pareto-optimal channel configurations (given the current $\boldsymbol{\theta}, \boldsymbol{x}, y$) across different FLOPs uniformly. While equation (2) precisely defines our goal, solving $g(\cdot)$ and $\alpha(\cdot)$ can be intractable since the functions are usually highly non-convex with respect to channel configurations and do not have analytical gradient information that admits first-order optimization algorithms. To tackle these challenges, we propose to model both $f_{\text{CE}}$ and $f_{\text{FLOPs}}$ via Gaussian Processes (GPs) (Rasmussen, 2003). This allows $\alpha(\cdot)$ to be approximated via Bayesian Optimization (BO) (Srinivas et al., 2009) with low overhead, which in turn enables $g(\cdot)$ to be implemented using binary search with $f_{\text{FLOPs}} \circ \alpha$. Concretely, binary search is done by repeating the following: start with $\boldsymbol{\lambda}_t$, solve $\alpha(\boldsymbol{\lambda}_t)$ using BO, and decrease or increase $\boldsymbol{\lambda}_{t+1}$ depending on the FLOPs of the current architecture $f_{\text{FLOPs}}(\alpha(\boldsymbol{\lambda}))$.

---

**Algorithm 1: PareCO**

**Input** : Model parameters $\boldsymbol{\theta}$, lower bound for width-multipliers $w_0 \in [0, 1]$, number of full iterations $F$, number of gradient descent updates $n$, number of $\boldsymbol{\lambda}$ samples $M$
**Output** : Trained parameter $\boldsymbol{\theta}$, approximate Pareto front $\mathcal{N}$

1   $\mathcal{H} = \{\}$     (*Historical minimizers $\widehat{\boldsymbol{c}}$*)
2   **for** *i = 1...F* **do**
3      $\boldsymbol{x}, y$ = sample_data()
4      $\boldsymbol{f}_{\text{CE}}, \boldsymbol{f}_{\text{FLOPs}} = f_{\text{CE}}(\mathcal{H}; \boldsymbol{\theta}, \boldsymbol{x}, y), f_{\text{FLOPs}}(\mathcal{H})$   (*Calculate the objectives for each $\widehat{\boldsymbol{c}} \in \mathcal{H}$*)
5      $\boldsymbol{g}$ = BuildGP-UCB( $\mathcal{H}, \boldsymbol{f}_{\text{CE}}, \boldsymbol{f}_{\text{FLOPs}}$ )     (*Build acquisition func. via BoTorch (Balandat et al., 2019)*)
6      widths = []
7      **for** *m = 1...M* **do**
8         $\widehat{\boldsymbol{c}}, \mathcal{N}$ = BOBS( $\boldsymbol{g}, \mathcal{H}, \boldsymbol{f}_{\text{CE}}, \boldsymbol{f}_{\text{FLOPs}}$ )     (*Algorithm 2*)
9         widths.append($\widehat{\boldsymbol{c}}$)
10     **end**
11     $\mathcal{H} = \mathcal{H} \cup$ widths         (*update historical data*)
12     widths.append($\boldsymbol{w_0}$)         (*smallest width for the sandwich rule in (Yu & Huang, 2019b)*)
13     **for** *j = 1...n* **do**
14         SlimmableTraining( $\boldsymbol{\theta}$, widths )   (*line 3-16 of Algorithm 1 in (Yu & Huang, 2019b)*)
15     **end**
16     $\mathcal{N}$=nonDominatedSort($\mathcal{H}, \boldsymbol{f}_{\text{CE}}, \boldsymbol{f}_{\text{FLOPs}}$)
17 **end**

---

### 3.2 APPROXIMATION VIA MULTI-OBJECTIVE BAYESIAN OPTIMIZATION

The goal is to approximate $\alpha(\cdot)$ and $g(\cdot)$ quickly with good quality. In this realm, Bayesian Optimization is known to be sample-efficient for doing so. Specifically, we are hoping to solve $\arg\min_{\boldsymbol{c}} \mathcal{T}_{\boldsymbol{\lambda}}(\boldsymbol{c}, \boldsymbol{\theta}, \boldsymbol{x}, y)$ efficiently using Bayesian Optimization where $\boldsymbol{c}$ here denotes the channel configurations. Instead of modeling $\mathcal{T}_{\boldsymbol{\lambda}}$ directly, we propose to model $f_{\text{CE}}$ and $f_{\text{FLOPs}}$ using two GPs. This design choice is mainly based on the fact that we would like to solve $g(\cdot)$ using binary search. More specifically, by having two GPs, the GPs can be reused across different $\boldsymbol{\lambda}$. On the other hand, if we directly model $\mathcal{T}_{\boldsymbol{\lambda}}$, we have to fit a GP for every $\boldsymbol{\lambda}$, which can be costly.

Having a separate GP for each objective of interest turns out to have no regret for optimization if a proper acquisition function is adopted (Paria et al., 2019). As a result, we follow Paria et al. (2019) and adopt the Upper Confidence Bound (UCB) (Srinivas et al., 2009) acquisition function for optimizing the problem of interest. To optimize the target objective, Bayesian Optimization proceeds sequentially. To begin, we start with training data $\mathcal{H}_{t-1} = \{\widetilde{\boldsymbol{c_1}}, \ldots, \widetilde{\boldsymbol{c_{t+o}}}\}$ and their function responses $f_i(\widetilde{\boldsymbol{c_\ell}}) \, \forall \, i \in \{\text{CE}, \text{FLOPs}\}, \ell \in [t + o]$ where $t$ denotes the current timestamp in the sequential optimization and $o$ is the initial size of the training data. Then, GPs are fitted for both objectives and acquisition functions for both GPs are combined using the augmented Tchebyshev scalarization with a specified $\boldsymbol{\lambda}$, which is then optimized to obtain the next point $\boldsymbol{c_{t+o+1}}$. This minimization is tractable because it minimizes the surrogate function instead of the unknown function. Under properly set hyperparameters for UCB, it is known that this procedure introduces no regret (Paria et al., 2019). In other words, if we allocate enough time for BO, *i.e.,*

$t \to \infty$, and set the output of BO to be $\widehat{c} \stackrel{\text{def}}{=} \widetilde{c_{t+o}}$, the method provides a close approximation, *i.e.*, $\mathcal{T}_{\boldsymbol{\lambda}}(\widehat{c}, \boldsymbol{\theta}, \boldsymbol{x}, y) \approx \min_{\boldsymbol{c}} \mathcal{T}_{\boldsymbol{\lambda}}(\boldsymbol{c}, \boldsymbol{\theta}, \boldsymbol{x}, y)$.

### 3.3 Approximation via temporal similarity

Equipped with BO, solving equation (2) can be done by running the following three steps iteratively: (1) sample $M$ target FLOPs, (2) solve the corresponding channel configurations using BO with binary search, and (3) perform a step of stochastic gradient descent. While BO provides a tractable mean for us to solve for Pareto-optimal channel configurations, it is still computationally expensive to run BO for every iteration of stochastic gradient descent. Thus, we propose to exploit temporal similarity for making the optimization practical. Specifically, we notice that $\boldsymbol{\theta}$ would not be drastically different across a few iterations since gradient descent itself relies on first-order Taylor approximation. As a result, we propose the following two approximations. First, instead of having many sequential queries in one subroutine of Bayesian optimization, we only perform one query and store the query to $\mathcal{H}$ for future Bayesian optimization. Note that for each query in BO, the cross-entropy loss will be reevaluated for each $\boldsymbol{c} \in \mathcal{H}$ to build faithful Gaussian Processes. Our second approximation is to perform $n$ SGD updates as opposed to one before another query for Bayesian optimization. We further provide theoretical analysis for approximation via temporal similarity in Appendix E.

**PareCO**   Based on this preamble, we present our algorithm, PareCO, in Algorithm 1. The proposed algorithm is Pareto-aware as the derivation stems from minimizing area under the Pareto curve. In short, PareCO has three steps: (1) build surrogate functions (*i.e.*, GPs) and acquisition functions (*i.e.*, UCBs) using historical data $\mathcal{H}$ and their function responses, (2) sample $M$ target FLOPs and solve for the corresponding widths (*i.e.*, $\widehat{c}$) via binary search with one query of BO, and (3) perform $n$ gradient descent steps using the solved widths. One can recover slimmable training (Yu & Huang, 2019b) by replacing lines 8 with randomly sampling a single width-multiplier for all the layers and setting $n = 1$ in line 13. The outputs of PareCO are both the shared weights and the Pareto-optimal widths. To obtain the Pareto-optimal widths, we use non-dominated sort based on the training loss and FLOPs for $c \in \mathcal{H}$.

---

**Algorithm 2:** Bayesian Optimization with Binary Search (BOBS)

**Input**   : Acquisition functions $\boldsymbol{g}$, historical data $\mathcal{H}$, $\boldsymbol{f}_{\text{CE}}$, $\boldsymbol{f}_{\text{FLOPs}}$, search precision $\epsilon$
**Output**: channel configurations $\widehat{c}$

1   $\beta = 10^{-6}$    (*A small positive number according to (Nakayama et al., 2009)*)
2   $\tilde{f}_{\text{FLOPs}} = \text{Uniform}(f_{\text{FLOPs,min}}, f_{\text{FLOPs,max}})$      (*Sample a target FLOPs*)
3   $\lambda_{\text{FLOPs}}, \lambda_{\text{min}}, \lambda_{\text{max}} = 0.5, 0, 1$
4   **while** $\left| \frac{f_{FLOPs}(\widehat{c}) - \tilde{f}_{FLOPs}}{FullModelFLOPs} \right| > \epsilon$ **do**               `// binary search`
5      $\widehat{c} = \arg\min_{\boldsymbol{c}} \left[ \max_{i \in \{\text{CE,FLOPs}\}} \lambda_i(g_i(\boldsymbol{c}) - \bar{g}_i) + \beta \sum_{i \in \{\text{CE,FLOPs}\}} \lambda_i g_i(\boldsymbol{c}) \right]$
6      **if** $f_{FLOPs}(\widehat{c}) > \tilde{f}_{FLOPs}$ **then**
7          $\lambda_{\text{min}} = \lambda_{\text{FLOPs}}$
8          $\lambda_{\text{FLOPs}} = (\lambda_{\text{FLOPs}} + \lambda_{\text{max}})/2$
9      **else**
10         $\lambda_{\text{max}} = \lambda_{\text{FLOPs}}$
11         $\lambda_{\text{FLOPs}} = (\lambda_{\text{FLOPs}} + \lambda_{\text{min}})/2$
12      **end**
13 **end**

---

## 4 Experiments

### 4.1 Performance gains introduced by PareCO

For all the PareCO experiments in this sub-section, we set $n$ such that PareCO only visits 1000 width configurations throughout the entire training ($|\mathcal{H}| = 1000$). Also, we set $M$ to be 2, which follows the conventional slimmable training method (Yu & Huang, 2019b) that samples two width configurations in between the largest and the smallest widths. As for binary search, we conduct at most 10 binary searches with $\epsilon$ set to 0.02, which means that the binary search terminates if the FLOPs difference is

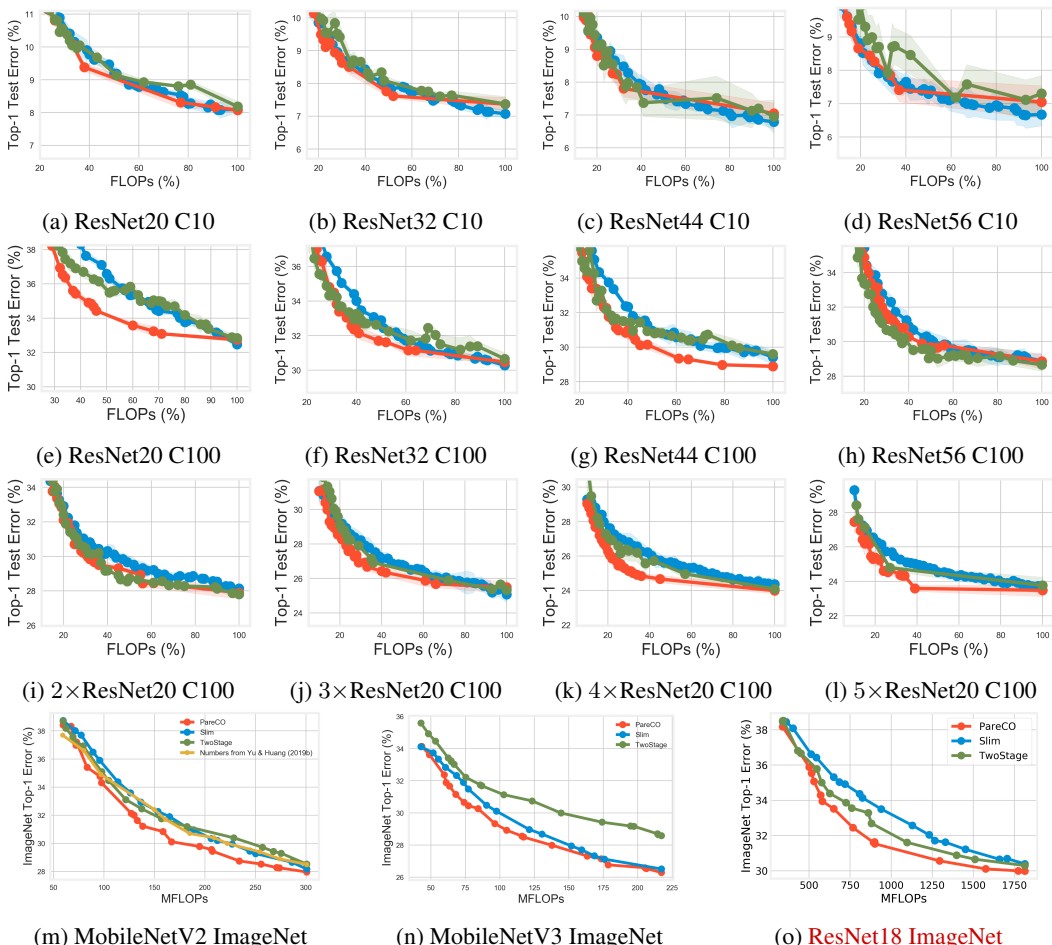

Figure 2: Comparisons among PareCO, Slim, and TwoStage. C10 and C100 denote CIFAR-10/100. For the CIFAR dataset, we perform three trials for each method and plot the mean and standard deviation. PareCO is better or comparable to Slim. The numerical results for ImageNet are detailed in Table 1 in Appendix G.

within a two percent margin relative to the full model FLOPs. On average, the procedure terminates by using 3.4 binary searches for results on ImageNet. The dimension of $c$ is network-dependent and is specified in Appendix B and the training hyperparameters are detailed in Appendix D. To arrive at the final set of architectures for PareCO, we use non-dominated sort based on the training loss and FLOPs for $c \in \mathcal{H}$.

We consider three datasets: CIFAR-10, CIFAR-100, and ImageNet. To provide informative comparisons, we verify our implementation for the conventional slimmable training with the reported numbers in (Yu & Huang, 2019b) using MobileNetV2 on ImageNet. Our results follow closely to the reported numbers as shown in Figure 2m, which makes our comparisons on other datasets convincing.

We compare to the following baselines:

- **Slim**: the conventional slimmable training method (the universally slimmable networks by Yu & Huang (2019b)). We select 40 architectures uniformly distributed across FLOPs and run a non-dominated sort using training loss and FLOPs to arrive at the points to be plotted.

- **TwoStage**: disjoint optimization that first trains the model with weight sharing, then uses search methods to find architectures that work well given the trained weights (similar to OFA (Cai et al., 2020) and BigNAS (Yu et al., 2020)). To compare fairly with PareCO,

we use multi-objective Bayesian optimization for the search. After optimization, we run a non-dominated sort for all the visited architectures $\mathcal{H}$ using training loss and FLOPs to arrive at the points to be plotted.

Compared to *Slim*, the proposed PareCO has demonstrated much better results across various networks and datasets. This suggests that channel optimization can indeed improve the efficiency of slimmable networks. Compared to *TwoStage*, PareCO is better or comparable across networks and datasets. This suggests that training network weights by uniformly sampling architectures regardless of Pareto-efficiency can be sub-optimal. More importantly, we find that such a training techniques result in low performances for all the sub-networks in MobileNetV3, which is potentially due to conflicts in the optimization process with a large number of sub-networks and small network capacity. From the perspective of training overhead, PareCO introduced minor overhead compared to Slim due to the temporal similarity approximation. More specifically, on ImageNet, PareCO incurs approximately 20% extra overhead compared to Slim.

Note that the performance among these three methods are similar for the CIFAR-10 dataset. This is plausible since when a network is more over-parameterized, there are many solutions to the optimization problem and it is easier to find solutions with the constraints imposed by weight sharing. In contrast, when the network is relatively less over-parameterized, compromises have to be made due to the constraints imposed by weight sharing. In such scenarios, PareCO outperforms Slim significantly, as it can be seen in CIFAR-100 and ImageNet experiments. We conjecture that this is because PareCO introduces a new optimization variable (width-multipliers), which allows better compromises to be attained. Along similar lines, from the experiments with ResNets on CIFAR-100 (Figure 2e to Figure 2h), we find that shallower models tend to benefit more from joint channel and weight optimization than their deeper counterparts.

As FLOPs may not necessarily reflect latency improvements since FLOP does not capture memory accesses, we in addition plot latency-*vs.*-error for the data in Figure 2m in Figure 3. The latency is measured on a single V100 GPU using a batch size of 128. When visualized in latency, PareCO still performs favorably compared to Slim and TwoStage for MobileNetV2 on ImageNet.

Lastly, to demonstrate the generality of PareCO, we consider another cost objective that is critical for on-device machine learning, *i.e.*, inference memory footprint (Yu et al., 2019). Inference memory footprint decides whether a model is executable or not on memory-constrained devices. We detailed the memory footprint calculation in Appendix F. Since PareCO is general, we can replace the FLOPs calculation with memory footprint calculation to optimize for memory-*vs.*-error. As shown in Figure 4, PareCO significantly outperform other alternatives. Notably, PareCO outperforms Slim by up to 8% top-1 accuracy for MobileNetV2. Such a drastic improvement comes from the fact that memory footpring depends only on the largest layer. As a result, slimming all the layers equally to arrive at networks with smaller memory footprint (as done in Slim) is less than ideal since only one layer contributes to the reduced memory.

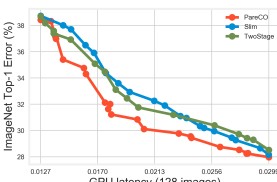 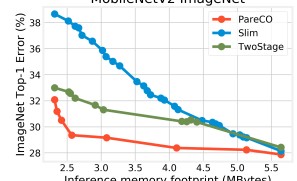 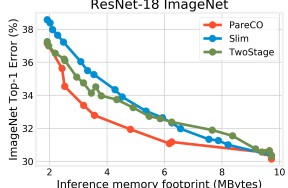

Figure 3: A latency-*vs.*-error view of Figure 2m.

Figure 4: Prediction error *vs.* inference memory footprint for MobileNetV2 and ResNet-18 on ImageNet.

### 4.2 ABLATION STUDIES

In this subsection, we ablate the hyperparameters that are specific to PareCO to understand their impact. We use ResNet20 and CIFAR-100 for the ablation with the results summarized in Figure 5.

**Binary search** Without binary search, one can also consider sampling $\boldsymbol{\lambda}$ uniformly from the $\Delta^{K-1}$, which does not require any binary search and is easy to implement. However, the issue with this

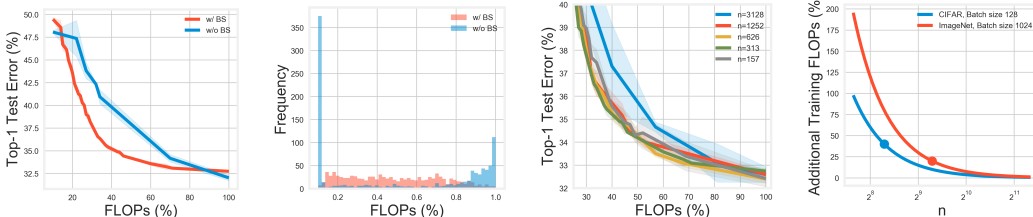

(a) Impact of binary search (BS).

(b) Histogram of FLOPs for $\mathcal{H}$ w/ and w/o BS.

(c) Performance for different $n$.

(d) Additional overhead over Slim for different $n$.

Figure 5: Ablation study for binary search and the number of gradient descent updates per full iteration using ResNet20 and CIFAR-100. Experiments are conducted three times and we plot the mean and standard deviation.

sampling strategy is that uniform sampling $\lambda$ does not necessarily imply uniform sampling in the objective space, *e.g.*, FLOPs. As shown in Figure 5a and Figure 5b, sampling directly in the $\lambda$ space results in non-uniform FLOPs and worse performance compared to binary search.

**Number of samples in BO**   We reduce the number of samples in BO by increasing the number of gradient descent updates. In previous experiments, we have $n = 313$, which results in $|\mathcal{H}| = 1000$. Here, we ablate $n$ to $156, 626, 1252, 3128$ such that $|\mathcal{H}| = 2000, 500, 250, 100$, respectively. With larger $n$, the algorithm introduce a worse approximation since there are overall less iterations put into Bayesian optimization. As shown in Figure 5c, we observe worse results with higher $n$. On the other hand, the improvement introduced by lower $n$ saturates quickly. The training overhead of PareCO as a function of $n$ compared to Slim is shown in Figure 5d where the dots are the employed $n$.

## 5   CONCLUSION

In this work, we propose to tackle the problem of training slimmable networks via a multi-objective optimization lens, which provides a novel and principled framework for optimizing slimmable networks. With this formulation, we propose a novel training algorithm, PareCO, which trains slimmable neural networks by jointly learning both channel configurations and the shared weights. In our empirical analysis, we extensively verify the effectiveness of PareCO over existing techniques on 15 dataset and network combinations and two types of cost objectives, *i.e.*, FLOPs and memory footprint. Our results highlight the potential of optimizing the channel counts for different layers jointly with the weights and demonstrate the power of such techniques for slimmable networks.

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

## A  PROOF FOR THEOREM 3.2

Let $\boldsymbol{\theta}$ be the network weights, $\mathcal{D}$ be the training data distribution, and $\boldsymbol{x}$ and $y$ be the training input and label. Furthermore, let $\alpha(\cdot)$ be a function that maps from $\boldsymbol{\lambda}$ to the corresponding Pareto channel configuration (*i.e.*, $\alpha(\boldsymbol{\lambda}) = \arg\min_{\boldsymbol{x}} \mathcal{T}_{\boldsymbol{\lambda}}(\boldsymbol{x})$), $g(\cdot)$ be a function that maps a certain FLOPs back to the corresponding $\boldsymbol{\lambda}$, (*i.e.*, $g = (f_{\text{FLOPs}} \circ \alpha)^{-1}$), and $f_{\text{CE}}$ be the cross entropy loss that takes channel configuration, weights and data, one can further capture the area under curve by Riemann integration over FLOPs:

$$
\begin{aligned}
A(\boldsymbol{\theta}, \boldsymbol{x}, y) &= \int_a^b f_{\text{CE}}(\alpha(\boldsymbol{g}(\boldsymbol{z})), \boldsymbol{\theta}, \boldsymbol{x}, y) dz \\
&\approx \sum_{i=0}^{r-1} f_{\text{CE}}(\alpha(\boldsymbol{g}(\boldsymbol{z_i})), \boldsymbol{\theta}, \boldsymbol{x}, y)(z_{i+1} - z_i),
\end{aligned}
\tag{3}
$$

where the approximation is done via Riemann sum with $r$ uniform partitions over FLOPs. As a result, our formal objective for optimizing a slimmable neural network is as follows:

$$
\arg\min_{\boldsymbol{\theta}} \mathbb{E}_{(\boldsymbol{x},y) \sim \mathcal{D}}[A(\boldsymbol{\theta}, \boldsymbol{x}, y)] \approx \arg\min_{\boldsymbol{\theta}} A(\boldsymbol{\theta}, \boldsymbol{x}, y)
\tag{4}
$$

$$
\approx \arg\min_{\boldsymbol{\theta}} \sum_{i=0}^{r-1} f_{\text{CE}}(\alpha(\boldsymbol{g}(\boldsymbol{z_i})), \boldsymbol{\theta}, \boldsymbol{x}, y)(z_{i+1} - z_i)
\tag{5}
$$

$$
= \arg\min_{\boldsymbol{\theta}} \sum_{i=0}^{r-1} f_{\text{CE}}(\alpha(\boldsymbol{g}(\boldsymbol{z_i})), \boldsymbol{\theta}, \boldsymbol{x}, y)
\tag{6}
$$

$$
= \arg\min_{\boldsymbol{\theta}} \frac{1}{r} \sum_{i=0}^{r-1} f_{\text{CE}}(\alpha(\boldsymbol{g}(\boldsymbol{z_i})), \boldsymbol{\theta}, \boldsymbol{x}, y)
\tag{7}
$$

$$
= \arg\min_{\boldsymbol{\theta}} \mathbb{E}_{z \sim [a,b]} f_{\text{CE}}(\alpha(\boldsymbol{g}(\boldsymbol{z})), \boldsymbol{\theta}, \boldsymbol{x}, y
\tag{8}
$$

$$
\approx \arg\min_{\boldsymbol{\theta}} \frac{1}{M} \sum_{m=1}^{M} f_{\text{CE}}(\alpha(\boldsymbol{g}(\boldsymbol{z}^{(m)})), \boldsymbol{\theta}, \boldsymbol{x}, y),
\tag{9}
$$

where equation 4 and equation 9 are done via Monte Carlo sampling and equation 5 is by equation 3. As for equation 6, we make use that $(z_{i+1} - z_i)$ is a positive constant across $i$, which does not affect $\arg\min$. Likewise, we can multiply a positive constant $\frac{1}{r}$ to arrive at equation 7.

## B  WIDTH PARAMETERIZATION

For ResNets with CIFAR, $\boldsymbol{c}$ has six dimensions and is denoted by $\boldsymbol{c}_{1:6} \in [0.316, 1]$, *i.e.*, one parameter for each stage and one for each residual connected layers in three stages. More specifically, the network is divided into three stages according to the output resolution, and as a result, there are three stages for all the ResNets designed for CIFAR. For example, in ResNet20, there are 7, 6, and 6 layers for each of the stages, respectively. Also, the layers that are added together via residual connection have to share the same width-multiplier, which results in one width-multiplier per stage for the layers that are connected via residual connections.

For MobileNetV2, $\boldsymbol{c}_{1:25} \in [0.42, 1]$, and therefore there is one dimension for each independent convolutional layer. Note that while there are in total 52 convolutional layers in MobileNetV2, not all of them can be altered independently. More specifically, for layers that are added together via residual connection, their widths should be identical. Similarly, the depth-wise convolutional layer should have the same width as its preceding point-wise convolutional layers. The same logic applies to MobileNetV3, which has 47 convolutional layers (excluding squeeze-and-excitation layers) and $\boldsymbol{c}_{1:22} \in [0.42, 1]$. In MobileNetV3, there are squeeze-and-excitation (SE) layers and we do not alter the width for the expansion layer in the SE layer. The output width of the SE layer is set to be the same as that of the convolutional layer where the SE layer is applied to. Note that there is no concept of expansion ratio for the inverted residual block in MobileNets in our width optimization. More specifically, the convolutional layer that acts upon expansion ratio is in itself just a convolutional layer

with tunable width. Also, we do not quantize the width to be multiples of 8 as adopted in the previous work (Sandler et al., 2018; Yu & Huang, 2019b). Due to these reasons, our $0.42\times$ MobileNetV2 has 59 MFLOPs, which has the same FLOPs as the $0.35\times$ MobileNetV2 in (Yu & Huang, 2019b; Sandler et al., 2018).

## C  WIDTH DIFFERENCES

In Figure 6, we visualize the widths learned by PareCO and contrast them with Slim for MobileNetV2 and MobileNetV3. Note that both PareCO and Slim are slimmable networks with shared weights and from the top row to the bottom row represent three points on the trade-off curve for Figure 2m and Figure 2o.

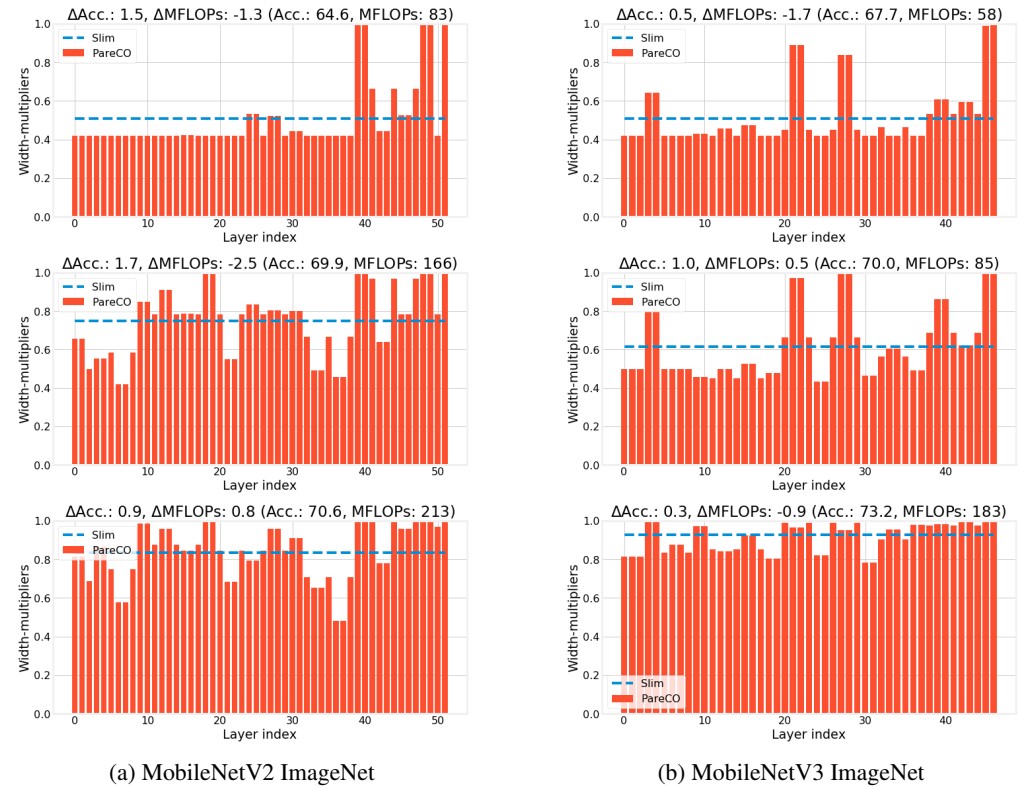

(a) MobileNetV2 ImageNet        (b) MobileNetV3 ImageNet

Figure 6: Comparing the width-multipliers between PareCO and Slim. The title for each plot denotes the relative differences (PareCO - Slim) and the numbers in the parenthesis are for PareCO.

## D  TRAINING HYPERPARAMETERS

We use PyTorch (Paszke et al., 2019) as our deep learning framework and we use BoTorch (Balandat et al., 2019) for the implementation of MOBO-RS, which works seamlessly with PyTorch. More specifically, for the covariance function of Gaussian Processes, we use the commonly adopted Matérn Kernel (Matérn, 2013) without changing the default hyperparameters provided in BoTorch. Similarly, we use the default hyperparameter provided in BoTorch for the Upper Confidence Bound acquisition function. To perform the optimization of line 6 in Algorithm 2, we make use of the API "*optimize_acqf*" provided in BoTorch. As a reference, with a single 1080Ti GPU, one can train a PareCO-ResNet20 on CIFAR-100 with around 3 hours. On the other hand, with 8 V100 GPUs on a single machine, one can train a PareCO-ResNet18 on ImageNet with 19 hours.

**CIFAR**  The training hyperparameters for the independent models are 0.1 initial learning rate, 200 training epochs, 0.0005 weight decay, 128 batch size, SGD with nesterov momentum, and cosine

learning rate decay. The accuracy on the validation set is reported using the model at the final epoch. For slimmable training, we keep the same exact hyperparameters but train $2\times$ longer compared to independent models, *i.e.*, 400 epochs.

**ImageNet**  Our training hyperparameters follow that of (Yu & Huang, 2019b). Specifically, we use initial learning rate of 0.5 with 5 epochs linear warmup (from 0 to 0.5), linear learning rate decay (from 0.5 to 0), 250 epochs, $4e^{-5}$ weight decay, 0.1 label smoothing, and we use SGD with 0.9 nesterov momentum. We use a batch size of 1024. For data augmentation, we use the "RandomResizedCrop" and "RandomHorizontalFlip" APIs in PyTorch. For MobileNetV2 we follow (Yu & Huang, 2019b) and use random scale between 0.25 to 1. For MobileNetV3, we use the default scale parameters, *i.e.*, from 0.08 to 1. The input resolution we use is 224. Besides scaling and horizontal flip, we follow (Yu & Huang, 2019b) and use color and lighting jitters data augmentataion with parameter of 0.4 for brightness, contrast, and saturation; and 0.1 for lighting. These augmentations can be found in the official repository of (Yu & Huang, 2019b)[2]. The entire training is done using 8 NVIDIA V100 GPUs.

## E  THEORETICAL ANALYSIS FOR TEMPORAL APPROXIMATION

The intuition behind the proposed approximation in Section 3.3 is the similarity for $\boldsymbol{\theta}$ across training iterations. In an extreme case, if we hold $\boldsymbol{\theta}$ constant throughout the training procedure, the approximation is equivalent to the original multi-objective BO. With that said, $\boldsymbol{\theta}$ changes gradually throughout training. To proceed with further theoretical understanding, we assume the network $f(\boldsymbol{x}, \boldsymbol{\theta})$ is L-Lipschitz. More formally,

$$f(\boldsymbol{x}, \boldsymbol{\theta}^t) - f(\boldsymbol{x}, \boldsymbol{\theta}^{t+1}) \leq L\|\boldsymbol{\theta}^t - \boldsymbol{\theta}^{t+1}\|_1, \forall \boldsymbol{\theta}^t, \boldsymbol{\theta}^{t+1}, \boldsymbol{x}. \tag{10}$$

Now, consider using stochastic gradient descent to update the weights $\boldsymbol{\theta}$, *i.e.*, $\boldsymbol{\theta}^{t+1} = \boldsymbol{\theta}^t - \alpha^t \boldsymbol{g}^t$ where $\boldsymbol{g}^t$ is the gradient of loss with respect to the weights and $\alpha^t$ is the learning rate at iteration $t$. Since $f$ is L-Lipschitz, we have $\|\boldsymbol{g}\|_1 \leq L$. Assuming using an exponential decaying learning rate with a factor $\gamma < 1$, we can further upper bound the functional differences across $n$ iterations of gradient descents as follows:

$$f(\boldsymbol{x}, \boldsymbol{\theta}^t) - f(\boldsymbol{x}, \boldsymbol{\theta}^{t+n}) \leq \sum_{i=t}^{t+n} \alpha^i \|g^i\| \leq n\alpha^t L. \tag{11}$$

Aligning with our intuition, the analysis reveals that larger $n$ implies poorer approximation. In multi-objective Bayesian optimization (Paria et al., 2019), the hyperparameter is searched over stationary objectives. In our case, due to temporal approximation, our cross entropy changes over time and the change is upper-bounded by $n\alpha^t L$. As a result, we can plug such an upper bound in the regret bound analysis of Bayesian optimization (Paria et al., 2019) to understand how $n$, $\alpha$, and $\gamma$ affect the optimality of Bayesian optimization. Specifically, we upper bound $f(\boldsymbol{x}, \boldsymbol{\theta}^t)$ with $f(\boldsymbol{x}, \boldsymbol{\theta}^{t+n}) + n\alpha^t L$ and use it in Lemma 2 and Lemma 3 from Paria et al. (2019) in Appendix B.1. With such a technique, a regret bound will have the following overhead in addition to the original regret bound in equation (14) of Paria et al. (2019):

$$\frac{2\alpha^1}{1-\gamma} n L \mathbb{E}[L_{\boldsymbol{\lambda}}] K, \tag{12}$$

where we have utilized the geometric progression of the exponential learning rate decay. In other words, without a decaying learning rate, the overhead can be unbounded. This analysis reveals that larger learning rate $\alpha^1$ and $n$ results in a worse regret bound.

## F  INFERENCE MEMORY FOOTPRINT CALCULATION

To demonstrate the generality of proposed PareCO, we in addition consider optimizing for the trade-off curve between prediction error and inference memory footprint. The inference memory footprint

---

[2]`https://github.com/JiahuiYu/slimmable_networks/blob/master/train.py#L43`

is a critical factor when it comes to deploying deep CNNs onto resource-constrained devices such as mobile phones or micro-controllers as motivated in the original slimmable neural network paper (Yu et al., 2019). We use a single image per batch to calculate the memory footprint. Specifically the inference memory footprint is characterized as follows:

$$
\begin{aligned}
FM_{in}^l &= W_{in}^l \times H_{in}^l \times C_{in}^l \\
FM_{out}^l &= W_{out}^l \times H_{out}^l \times C_{out}^l \\
\text{Weights}^l &= K_w^l \times K_h^l \times C_{in}^l \times C_{out}^l / G^l \\
\text{Skip}^l &= W_{out}^l \times H_{out}^l \times C_{skip}^l \\
MEM &= \max_l \left( FM_{in}^l + FM_{out}^l + \text{Weights}^l + \text{Skip}^l \right),
\end{aligned}
\tag{13}
$$

where $FM_{in}^l$ and $FM_{out}^l$ denote the input and output feature map sizes of layer $l$, Weights$^l$ denotes the size of the weights in layer $l$, and Skip$^l$ denotes the memory cost of storing the feature maps from skip connections. $W$ and $H$ represent the width and height of the feature map. $K_w$ and $K_h$ denote the kernel size. Lastly, $C_{in}$, $C_{out}$ and $G$ denote the input channel, output channel, and the number of groups for convolutional layer $l$.

## G   NUMERICAL RESULTS FOR IMAGENET

We summarize the numbers from Figure 2m and Figure 2o in Table 1 for future work to compare easily.

| | MobileNetV2 | | | | MobileNetV3 | | |
|---|---|---|---|---|---|---|---|
| MFLOPs | Independently trained (Sandler et al., 2018) | Slim | PareCO | MFLOPs | Independently trained (Howard et al., 2019) | Slim | PareCO |
| 59 | 60.3 | 61.4 | **61.5** (+0.1) | 40 | **64.2** | - | - |
| 71 | - | 61.9 | **63.0** (+1.1) | 42 | - | 65.8 | **65.9** (+0.1) |
| 84 | - | 63.0 | **64.6** (+1.6) | 51 | - | 66.3 | **66.6** (+0.3) |
| 95 | - | 64.0 | **65.1** (+1.1) | 60 | - | 67.2 | **67.7** (+0.5) |
| 97 | 65.4 | - | - | 69 | **68.8** | - | - |
| 102 | - | 64.7 | **65.5** (+0.8) | 73 | - | 68.1 | **68.8** (+0.7) |
| 136 | - | 67.1 | **68.2** (+1.1) | 84 | - | 69.0 | **70.0** (+1.0) |
| 149 | - | 67.6 | **69.1** (+1.5) | 118 | - | 71.0 | **71.4** (+0.4) |
| 169 | - | 68.2 | **69.9** (+1.7) | 121 | - | 71.0 | **71.6** (+0.6) |
| 209 | 69.8 | - | - | 155 | **73.3** | - | - |
| 212 | - | 69.7 | **70.6** (+0.9) | 168 | - | 72.7 | **72.8** (+0.1) |
| 244 | - | 70.5 | **71.0** (+0.5) | 183 | - | 73.0 | **73.2** (+0.2) |
| 300 | 71.8 | 72.0 | **72.1** (+0.1) | 217 | **75.2** | 73.5 | 73.7 (+0.2) |

Table 1: MobileNetV2 and MobileNetV3 on ImageNet. The number in the parenthesis for PareCO are the improvements compared to the corresponding Slim. Bold represents the highest accuracy of a given FLOPs.

