# OpenReview forum: "PareCO: Pareto-aware Channel Optimization for Slimmable Neural Networks"
_ICLR.cc/2021/Conference — Reject_

### Official Review · AnonReviewer1 · 2020-10-23
**A joint optimization approach with rigorous theoritical support; concerns about evidence and reproducing**

**Rating:** 6
**Confidence:** 4

**Review:**

This paper proposes a multi-objective optimization approach to jointly train both channel configurations and shared weights of slimmable networks. It is theoretically proven that by minimizing the cross entropy loss for the Pareto-optimal channel configurations, the joint optimization can be approximately accomplished. Based on the objective, this paper proposes several approximation algorithms and sampling strategies to obtain a target network with better trade-off curve.

Experiments are thoroughly conducted on a range of network settings and datasets. The results experimentally show that the channel optimization can improve the performance and efficiency of slimmable networks. This paper is logically organized and the motivation is based on sufficient theoretical supports.

A few constructive criticisms or concerns as follows:
1.	The approximation algorithms and theorem 3.2 in this paper points out several ways to approximate the joint-learning objective. However, the authors did not provide descriptions about the intuition or advantages of the proposed approximation methods. Detailed theoretical analysis on the discrepancy between approximation and the ideal objective had better to be provided.
2.	Experiments cannot directly support the authors’ conclusion “less over-parameterized networks benefit more from joint channel and weight optimization” because the experiments cannot eliminate the impact of other variables.
3.	It might be difficult to reproduce this research and the experiment results since the search space is huge and the authors did not report the search cost (e.g. GPU-days and memory cost).

---

> ### Author Response · Authors · 2020-11-19
> **responses to r1**
>
> We thank the reviewer for their feedback and finding our proposed method theoretically grounded with thorough experiments. Let us try to address the raised concerns.
>
> > R1: “The approximation algorithms and theorem 3.2 in this paper points out several ways to approximate the joint-learning objective. However, the authors did not provide descriptions about the intuition or advantages of the proposed approximation methods. Detailed theoretical analysis on the discrepancy between approximation and the ideal objective had better to be provided.”
>
> As mentioned in Section 3.3, the intuition behind the approximation comes from temporal similarity for the underlying $\theta$ across training iterations. In an extreme case, if we hold $\theta$ constant throughout the procedure, the approximation is equivalent to the original multi-objective BO. That being said, $\theta$ does change gradually throughout the optimization. Inspired by the reviewer, we have provided additional theoretical analysis on how the proposed approximation technique affects the regret bound for Bayesian optimization in Appendix E.
>
> > R1: “Experiments cannot directly support the authors’ conclusion “less over-parameterized networks benefit more from joint channel and weight optimization” because the experiments cannot eliminate the impact of other variables.”
>
> We thank the reviewer for raising a good point. We have indeed made a broader claim than what our current empirical results support. To make the claim more precise, we will use the following instead, “From the experiments with ResNets on CIFAR-100 (Fig 2(e-h)), we find that shallower models tend to benefit more from joint channel and weight optimization than their deeper counterparts.”
>
> > R1: “It might be difficult to reproduce this research and the experiment results since the search space is huge and the authors did not report the search cost (e.g. GPU-days and memory cost).”
>
> To better improve the reproducibility, we have anonymously open-sourced our code for this project at (https://github.com/iclr2021-pareco/anonymous-pareco). The search cost depends on the datasets. On CIFAR-100, it takes only 3 hours for ResNet-20 to run on a single 1080TI GPU. On ImageNet, it takes 19 hours on 8 V100 GPUs for training a PareCO-ResNet-18 (100 epochs). For MobileNetV2 with ImageNet, we follow the training hyperparameter in prior literature, which uses 250 training epochs. While it takes about 2 days to finish, this is due to the 250 epochs set in prior work. We have added the results for ResNet-18 on ImageNet for future work to compare and contrast with lower training overhead. We agree with the reviewer that such information can be useful and have added it in Appendix D along with the training hyperparameters.

---

### Official Review · AnonReviewer3 · 2020-10-27
**Good work, successfully extends previous work with proper techniques**

**Rating:** 6
**Confidence:** 3

**Review:**

The paper directly extends the “Slimmable network” by using per-layer width multipliers to allow more flexible network configurations. PareCO mainly aims to optimize both width multipliers and shared weights while considering Pareto-optimal. (Accuracy & Speed) Because the problem is intractable, the authors adopt the Bayesian optimization model of CE loss and FLOPs. To improve search speed, several methods (temporal similarity, binary search) are also introduced. Compared to previous work (SLIM), PareCO consistently achieves better Pareto-optimal performance.

The paper is clearly written and has some significance. Related works are well established. Additional techniques, such as storing query for BO, binary searching, restrict to 1000 configurations, makes the proposed algorithm practical. The experimental result seems quite promising.

In my opinion, the novelty of the paper is not heterogenous width multiplier nor supernet-like training. Many works state that per-layer channel sparsity is important, and the idea of per-layer width multiplier is not new. I think this paper has some novelty, especially the “Pareto-Aware” part, but is not significant.

Minor questions/suggestions:

1.	Several works in NAS target Pareto-Optimal architecture. Some use Bayesian optimization.  Consider adding them to the related works section... [Jin-Dong Dong, 2018] DPP-Net, [Jin-Dong Dong, 2018] PPP-Net, [An-Chieh Cheng, 2018] Searching Toward Pareto-Optimal Device-Aware Neural Architectures, [Md Shahriar Iqbal, 2020] FlexiBO, etc.

2.	Isn’t the CE re-evaluation part for every query in history too costly? Especially for a big dataset like ImageNet. Is this the reason for “20% extra overhead compared to Slim”?

---

> ### Author Response · Authors · 2020-11-19
> **responses to r3**
>
> We thank the reviewer for their feedback and for finding our paper clearly written with some significance. We appreciate the pointers and have incorporated the reviewer’s suggestions and included the provided references into the related work section.
>
> > R3: "In my opinion, the novelty of the paper is not heterogenous width multiplier nor supernet-like training. Many works state that per-layer channel sparsity is important, and the idea of per-layer width multiplier is not new. I think this paper has some novelty, especially the “Pareto-Aware” part, but is not significant."
>
> We would like to summarize the novelty and the significance of the proposed method below:
>
> - We proposed a method that incorporates multi-objective optimization into the training loop of weight-sharing networks by introducing novel techniques such as temporal similarity and binary search as acknowledged by the reviewer. We have additionally demonstrated the importance of these techniques in our ablation study (Section 4.2).
> - While multi-objective optimization has been used in NAS literature, it is relatively new when it comes to weight-sharing networks. To our best knowledge, the only methods on this front are BigNAS and OFA, which use the TwoStage method. However, as demonstrated in the paper, our proposed method outperforms such a TwoStage method.
> - To further demonstrate the significance of the proposed approach, we have conducted new experiments for optimizing the error vs. memory footprint trade-off curve in Section 4.1, which demonstrated the generality of the proposed method. The inference memory footprint is a crucial metric when it comes to deploying models to mobile devices as this determines whether a device is able to run a given model. Note that memory footprint is determined by the largest layer, which means that, to arrive at models with smaller memory footprint,  one really needs to judiciously reduce the channels from the largest layer as opposed to uniformly reducing all the layers. As can be seen from Figure 4, PareCO significantly outperforms other alternatives when it comes to error vs. inference memory footprint (**up to 8% top-1 accuracy improvements on ImageNet**). This further establishes the significance of the proposed method: generality and much better performance.
>
> > R3: “Isn’t the CE re-evaluation part for every query in history too costly? Especially for a big dataset like ImageNet.”
>
> Yes, it accounts for a large portion of the introduced overhead (extra 20% compared to normal simmable training). However, it only requires |H| number of network forwards as we use a single batch to evaluate the cross entropy loss for each architecture, which is a Monte Carlo sample as shown in equation (4). As a result, this overhead does not increase with the dataset sizes. Our particular setting is very useful for deployment scenarios where the inference compute varies, e.g., optimizing models for many different mobile platforms. In these scenarios, the training cost is paid only once while the optimized results can be re-used across varying inference compute budgets.

---

### Official Review · AnonReviewer4 · 2020-10-28
**PareCO: Pareto-aware Channel Optimization for Slimmable Neural Networks**

**Rating:** 5
**Confidence:** 4

**Review:**

This paper proposes a new multi-objective optimization method for slimmable neural networks and jointly optimizes network architecture and weights. However, the use of multi-objective optimization methods for networks is not new. The advantage of this paper over existing work in this regard is not clear. Besides, the key term “Pareto-aware” in the title is not clearly defined.

---

> ### Author Response · Authors · 2020-11-19
> **responses to r4**
>
> We appreciate the reviewer’s feedback. We have considered the suggestions and updated the paper to make it clear that “Pareto-aware” comes from the fact that our method is derived from maximizing Pareto-efficiency.
>
> > R4: “However, the use of multi-objective optimization methods for networks is not new. The advantage of this paper over existing work in this regard is not clear.”
>
> The closest approach of applying multi-objective optimization to weight-sharing networks is the TwoStage approach (BigNAS and OFA cited in the paper) mentioned in the third paragraph of Section 1. As demonstrated in Section 4, our novel method outperformed such baselines. More importantly, we apply the optimization in a novel way. Specifically, it is not clear how one should incorporate multi-objective optimization into the weight-sharing training. Prior art uses a TwoStage method, which is worse compared to our proposed method; this is because the training objective for TwoStage essentially requires all sub-networks to be good, which is not necessary. All one should care about is the Pareto-optimal ones. To reflect this requirement, we propose to incorporate multi-objective optimization into the training loop of the weight-sharing network. However, this is intractable, as shown in our derivation. Hence, we have proposed a novel sampling technique called Bayesian optimization with binary search (BOBS) and a novel temporal approximation to resolve the intractability challenge. During the rebuttal, we have provided additional results on error vs. Inference memory footprint in Section 4.1, which further show the significance and generality of the proposed approach over other alternatives.

---

### Official Review · AnonReviewer2 · 2020-10-30
**Lack of Novelty**

**Rating:** 4
**Confidence:** 4

**Review:**

This paper extends on the existing network slimming approach, computing a heterogenous width for each layer. The layer width is computed by solving a multi objective optimization problem based on pareto distribution.

Pros:

The paper is well motivated and well written.
The experiments are run across mutliple datasets and models. And also compared with existing approaches in the literature.
Cons:

The main aspect lacking in the paper is the novelty. In the three steps of the approach explained in "Pareco" section in Algorithm 1, the first and third step already exist in Network slimming approaches. And Step 2 - solving for 'c' is a multi objective optimization which directly exists in BoTorch library. So, I find lack of novelty and do not learn much or take away much reading this paper.

Also, by additionally solving for 'c', I do not realize what is the value add in terms of performance. In resnet-like architectures, there is no significant improvement in accuracy- as shown in Fig. 2 (a) to (l). The results in Mobilenet is slightly better as compared to NS - ~1 to 1.5% improvement, which is not significant, in my opinion. This is further established by interpreting, Fig 3(c) - for all different values of n (unless extremely high), we achieve a similar performance by the proposed approach.

Thus, I believe this paper is basically a hyperparameter tune over existing NS approaches, results in marginally not-so-sure significant improvement in results.

---

> ### Author Response · Authors · 2020-11-19
> **responses to r2 (cont.)**
>
> > R2: “The results in Mobilenet is slightly better as compared to NS - ~1 to 1.5% improvement, which is not significant, in my opinion.”
>
> We respectfully argue that 1% to 1.5% top-1 accuracy on ImageNet is significant. As evident from recent publications throughout recent ICLR and NeurIPS [2-9]. More specifically,
>
> [2] improves upon its prior art by 0.6% (74 - > 74.6)
>
> [3] improves upon the baseline by 0.3% (75.3 -> 75.6)
>
> [4] improves upon its prior art by 0.8% (72.5 -> 73.3)
>
> [5] improves upon the baseline by 0.97% (76.34 -> 77.31)
>
> [6] improves upon its prior art by 0.23% (74.95 -> 75.18)
>
> [7] improves upon its prior art by 1.49% (58.35 -> 59.84)
>
> [8] improves upon its prior art by 0.7% (75.5 -> 76.2)
>
> [9] improves upon its prior art by 1.5% (74.5 -> 76.0)
>
> Additionally, our improvements are acknowledged by R3: “The experimental result seems quite promising.”
>
> > R2: “This is further established by interpreting, Fig 3(c) - for all different values of n (unless extremely high), we achieve a similar performance by the proposed approach.”
>
> The n values control the level of temporal approximation and the tolerance to the approximation depends on both the network and the dataset. Fig. 3(c) shows that |H|=500 can be good enough while |H|=100 is not, due to large approximation error when it comes to ResNet20 and CIFAR-100. We respectfully argue that it is incorrect to interpret the results of Fig. 3(c) as “there is no value in terms of solving for c”. First, we can clearly see that |H|=500 is better than |H|=100, which says if we solve for c with higher quality, we obtain better results. Second, directly comparing PareCO with Slim shows the benefits of having c optimized. Moreover, we would like to mention that having good results with lower |H| is an advantage of PareCO rather than a disadvantage as the introduced overhead is inversely proportional to |H|.
>
> > R2: “Thus, I believe this paper is basically a hyperparameter tune over existing NS approaches”
>
> We would like to emphasize that this work is not about tuning the hyperparameter introduced in prior work to obtain better results. Instead, we propose to allow the width to vary heterogeneously across layers for a slimmable network. Additionally, we propose a novel and principled method for achieving such a goal. In fact, the proposed method is a generalization of the prior work, and not the other way around.
>
> [1] Yu, Jiahui, et al. "Slimmable neural networks." arXiv preprint arXiv:1812.08928 (2018).
> [2] Cai, Han, Ligeng Zhu, and Song Han. "Proxylessnas: Direct neural architecture search on target task and hardware." ICLR 2019.
> [3] Chen, Hao-Yun, et al. "Complement Objective Training." ICLR 2019.
> [4] Liu, Hanxiao, Karen Simonyan, and Yiming Yang. "Darts: Differentiable architecture search." ICLR 2019.
> [5] Chen, Chun-Fu, et al. "Big-little net: An efficient multi-scale feature representation for visual and speech recognition." ICLR 2019.
> [6] You, Zhonghui, et al. "Gate decorator: Global filter pruning method for accelerating deep convolutional neural networks." NeurIPS 2019.
> [7] Chen, Shangyu, et al. “MetaQuant: Learning to Quantize by Learning to Penetrate Non-differentiable Quantization.” NeurIPS 2019.
> [8] Dong, Xuanyi, and Yi Yang. "Network Pruning via Transformable Architecture Search." NeurIPS 2019.
> [9] Nayman, Niv, et al. "XNAS: Neural Architecture Search with Expert Advice." NeurIPS 2019.

---

> ### Author Response · Authors · 2020-11-19
> **responses to r2**
>
> We thank the reviewer for their feedback and for finding our paper well motivated and well written with extensive empirical results. Regarding the reviewer’s concern, we believe there are some misunderstandings which we are addressing below.
>
> > R2: “The main aspect lacking in the paper is the novelty.”
>
> Our work is the first work to jointly learn the channel counts and the weights for slimmable neural networks. Furthermore, the proposed method **is not** simply adopting multi-objective optimization using Bayesian optimization (MOBO).
>
> The key question is when and how to use MOBO. Following the NAS literature, we can apply MOBO after training a super-network, which is exactly our TwoStage baseline. As shown in Section 4, our proposed method outperforms such an alternative. On the other hand, one can try to apply MOBO during the training of the shared weights, however, it is not readily clear how to do so. A naive approach is to use MOBO to obtain the entire Pareto front for each iteration, which would be impractically costly. Our proposed method approaches this challenge from a principled derivation (Theorem 3.2), which includes a principled sampling strategy implemented with binary search (Algorithm 2) and a temporal approximation for BO (line 11 in Algorithm 1). Both aspects are technically novel and necessary for the proposed algorithm (as demonstrated in the ablation study in the current Figure 5: MOBO can perform much worse without sampling with binary search or with a long horizon in temporal approximation, i.e., large n). As a result, we respectfully argue that the fact that one can conveniently implement our algorithm with BoTorch should be an advantage when it comes to reproducibility as opposed to a drawback. In fact, without these derivations, it is not clear if we should solve for ‘c’ and with respect to what objective.
>
> > R2: “Also, by additionally solving for 'c', I do not realize what is the value add in terms of performance.”
>
> The intuition behind optimizing the widths (c values) for different layers is that **different layers have different costs and contribute to the final accuracy differently**. We can further demonstrate this phenomenon by looking at another cost: memory footprint as opposed to FLOPs. The inference memory footprint is an important aspect for on-device machine learning [1] and it is directly related to the largest layer of the network, which means that if we uniformly shrink all the channels (as done in Slimmable Networks, or Slim, as denoted on the paper) to arrive at a network with smaller memory footprint, only one particular layer contributes to the reduced memory footprint. Intuitively, a better strategy in this case is to only shrink the bottleneck layer to maintain the most representational power. All in all, due to the different impact that different layers have on the cost and accuracy, it is the best if we “optimize” channel allocation in a Pareto-optimal fashion. To this end, our proposed method does so, and  with a principled derivation. To empirically validate our aforementioned example, we have conducted new experiments on ImageNet with MobileNetV2 using PareCO to optimize for Accuracy vs. Memory footprint and compared to other alternatives. **The difference between PareCO and Slim can be as large as 8% top-1 accuracy on ImageNet (62.3% vs 70.6%)** as shown in Figure 4, which demonstrates the importance of solving for ‘c’. We have added these discussions to the current paper in Section 4.1.
>
> > R2: “In resnet-like architectures, there is no significant improvement in accuracy- as shown in Fig. 2 (a) to (l)”
>
> From Fig. 2(e) to (l), we can still observe significant performance improvement over the baseline Slim (red vs. blue). For example, in Fig. 2(e) at 70% FLOPs, PareCO achieves 33% top-1 error vs. 35% for Slim. That being  said, we think that the improvements depend on the target dataset as opposed to the network family. Inspired by the question raised by the reviewer, we have run extra experiments with ResNet-18 on ImageNet. As can be observed in Fig. 2(o), PareCO significantly outperforms other alternatives, which is similar to Fig. 2(m). Specifically, up to 2% top-1 error reduction can be achieved comparing PareCO to Slim using ResNet-18.

---

### Decision · Program_Chairs · 2021-01-07
**Final Decision**

**Decision:**

Reject

**Comment:**

The paper extends the work of “slimmable networks” in that it aims to find a single set of weights suitable for multiple FLOP/accuracy tradeoff (or memory/accuracy tradeoff). The main novelty of the paper is in adapting known techniques from bayesian optimization (BO) to the setting at hand, resulting in a modified training technique. The experiments show a performance lift when compared against the original slimmable networks, as well as other approaches called “two stage” that alternate between optimizing the weights and the architecture.
The paper provides a practical approach to an important problem yielding non-trivial results. The main weakness of the paper seems to be its novelty. Although it is not possible to naively apply the multi-objective optimization with NAS techniques, the reviews seem to indicate that the innovation required to do so is not sufficient to meet the ICLR bar. This is indeed a borderline case, but given the competing papers, my tendency is towards rejecting the paper.